# STRENGTH IN NUMBERS: TRADING-OFF ROBUSTNESS AND COMPUTATION VIA ADVERSARIALLY-TRAINED ENSEMBLES

## ABSTRACT

While deep learning has led to remarkable results on a number of challenging problems, researchers have discovered a vulnerability of neural networks in adversarial settings, where small but carefully chosen perturbations to the input can make the models produce extremely inaccurate outputs. This makes these models particularly unsuitable for safety-critical application domains (*e.g.* self-driving cars) where robustness is extremely important. Recent work has shown that augmenting training with adversarially generated data provides some degree of robustness against test-time attacks. In this paper we investigate how this approach scales as we increase the computational budget given to the defender. We show that increasing the number of parameters in adversarially-trained models increases their robustness, and in particular that ensembling smaller models while adversarially training the entire ensemble as a single model is a more efficient way of spending said budget than simply using a larger single model. Crucially, we show that it is the adversarial training of the ensemble, rather than the ensembling of adversarially trained models, which provides robustness.

## 1 INTRODUCTION

Deep neural networks have demonstrated state-of-the-art performance in a wide range of application domains Krizhevsky et al. (2012). However, researchers have discovered that deep networks are in some sense 'brittle', in that small changes to their inputs can result in wildly different outputs (Huang et al., 2017; Jia & Liang, 2017; Szegedy et al., 2013). For instance, practically imperceptible (to human) modifications to images can result in misclassification of the image with high confidence. Not only are networks susceptible to these 'attacks', but these attacks are also relatively easy to compute using standard optimization techniques (Carlini & Wagner, 2017b; Goodfellow et al., 2014). These changes are often referred to as *adversarial perturbations*, in the sense that an adversary could craft a very small change to the input in order to create an undesirable outcome. This phenomenon is not unique to image classification, nor to particular network architectures, nor to particular training algorithms (Papernot et al., 2016; 2017).

Adversarial attacks can be broken into different categories depending on how much knowledge of the underlying model the adversary has access to. In 'white-box' attacks the adversary has full access to the model, and can perform both forward and backwards passes (though not change the weights or logic of the network) (Carlini & Wagner, 2017a; Goodfellow et al., 2014). In the 'black-box' setting the adversary has no access to the model, but perhaps knows the dataset that the model was trained on (Papernot et al., 2016; 2017). Despite several recent papers demonstrating new defences against adversarial attacks (Akhtar & Mian, 2018; Guo et al., 2017; Liao et al., 2017; Song et al., 2017; Tramèr et al., 2018; Warde-Farley & Goodfellow, 2016; Xie et al., 2017; Yuan et al., 2017), recent papers have demonstrated that most of these new defences are still susceptible to attacks and largely just obfuscate the gradients that the attacker can follow, and that non-gradient based attacks are still effective Uesato et al. (2018); Athalye et al. (2018).

**Exploring Tradeoff of Computation and Robustness** In many safety-critical application domains (*e.g.* self-driving cars), robustness is extremely important even if it comes at the cost of increased

computation. This motivated the central question considered by this paper: *Is it possible to increase adversarial robustness of a classifier at the cost of increased computation?*

There are a number of possibilities to employ extra computation available at runtime. We can use a much larger model that requires more time to run, execute the original model multiple times and aggregate the predictions, or instead of using a single model, make predictions from a portfolio or ensemble of models. While researchers have proposed the use of portfolios and ensembles as a mechanism to improve adversarial robustness Abbasi & Gagné (2017); Thilo Strauss (2017), our experimental results indicate that stronger adversaries are able to attack the ensembles successfully.

**Contributions**   In this paper, we study and analyze the trade-off of adversarial robustness and computation (memory and runtime). We propose the use of adversarial training of ensemble of models and through an exhaustive ablative analysis make the following empirical findings:

- increased computation and/or model size can be used to increase robustness,
- ensembles on their own are not very robust, but can be made robust through adversarial training where the ensemble is treated as a single model,
- adversarially trained ensembles are more robust than adversarially trained individual models requiring the same amount of parameters/computation

**Related Work**   Recently, Tramèr et al. (2018) investigated the use of ensembles for adversarial robustness. However, their goal and approach was quite different from the technique we are investigating. In Tramèr et al. (2018), the authors generated adversarial perturbations using an ensemble of *pre-trained* models in order to transfer the example to another model during training. This procedure decouples adversarial example generation from the current model, and consequently the model being trained cannot simply 'overfit' to the procedure for generating adversarial examples, which they generally took to be single-step attack methods. The authors demonstrated strong robustness of the resulting trained model to black-box attacks. By contrast, in this paper we investigate using an ensemble of models as our predictive model, and we train the models using multi-step adversarial training. We show increased robustness to both black-box and white-box adversarial attacks using this strategy.

## 2   PRELIMINARIES

Here we lay out the basics of attacking a neural network by the generation of adversarial examples. Denote an input to the network as $x \in \mathbb{R}^d$ with correct label $\hat{y} \in \mathcal{Y} \subset \mathbb{N}$, and let $m_\theta : \mathbb{R}^d \to \mathbb{R}^{|\mathcal{Y}|}$ be the mapping performed by the neural network which is parameterized by $\theta \in \mathbb{R}^p$. Let $L : \mathcal{Y} \times \mathbb{R}^{|\mathcal{Y}|} \to \mathbb{R}$ denote the loss we are trying to minimize (*e.g.*, the cross-entropy). When training a neural network we seek to solve

$$\text{minimize} \quad \mathbb{E}_{(x,y) \sim \mathcal{D}} L(\hat{y}, m_\theta(x)) \tag{1}$$

over variable $\theta$, where $\mathcal{D}$ is the data distribution. Given any fixed $\theta$ we can generate (untargeted) adversarial inputs by perturbing the input $x$ so as to *maximize* the loss. We restrict ourselves to small perturbations around a nominal input, and we denote by $\mathcal{B}$ this set of allowable inputs. For example, if we restrict ourselves to small perturbations in $\ell_\infty$ norm around a nominal input $x^{\text{nom}}$ then we could set $\mathcal{B} = \{x \mid \|x - x^{\text{nom}}\|_\infty \le \epsilon\}$ where $\epsilon > 0$ is the tolerance. A common approach for generating adversarial examples is projected gradient descent Carlini & Wagner (2016), *i.e.*, to iteratively update the input $x$ by

$$\tilde{x}^{k+1} = \Pi_{\mathcal{B}}(\tilde{x}^k + \eta \nabla_x L(y, m_\theta(\tilde{x}^k))), \tag{2}$$

where typically $x^0 = x + \epsilon$ for some noise $\epsilon$, $\eta > 0$ is a step-size parameter and $\Pi_{\mathcal{B}}$ denotes the Euclidean projection on $\mathcal{B}$. We add noise to the initial point so that the network can't memorize the training dataset and mask or obfuscate the gradients at that point Uesato et al. (2018); Athalye et al. (2018), in other words the added noise encourages generalization of adversarial robustness to the test dataset. If instead of using the gradient we just use the sign of the gradient then this is the fast-gradient-sign method Goodfellow et al. (2014). Empirically speaking, for most networks just a few steps of either of these procedures is sufficient to generate an $\tilde{x}$ that is close to $x^{\text{nom}}$ but has a different label with high confidence.

In this paper we are primarily concerned with the performance of ensembles of models when trained with *adversarial training* Madry et al. (2017). In adversarial training we train a network to minimize a weighted sum of two losses (where the relative weighting is a hyper-parameter). The first loss is the standard loss of the problem we are trying to solve on the normal training data, *e.g.*, the cross-entropy for a classification task. The second loss is the same function as the first loss, except evaluated on *adversarially generated data*, where typically the adversarial data is generated by attacking the network at that time-step. In other words we replace the problem in eq. (1) with

$$\text{minimize} \quad \mathbb{E}_{(x,y)\sim\mathcal{D}}(L(\hat{y}, m_\theta(x)) + \rho L(\hat{y}, m_\theta(\tilde{x}))) \tag{3}$$

where $\rho \geq 0$ is the weighting parameter and $\tilde{x}$ is an adversarial example generated from $x$ at model parameters $\theta$ using, for example, the update in eq. (2). This problem is usually approximated by sampling and minimizing the empirical expectation.

## 3 ADVERSARIALLY-TRAINED ENSEMBLES

In this section we lay out the basic strategy of using ensembles of models to increase robustness to adversarial attacks. The notion of ensemble used here simply involves taking $k$ separately-parameterized models and averaging their predictions. If the output of network $i$ as a function of input $x$ and with network parameters $\theta_i$ is given by $p(\cdot|x, \theta_i) = m_{\theta_i}(x)$, then the output of the ensemble is

$$p(y|x) = \frac{1}{k}\sum_{i=1}^{k} p(y|x, \theta_i).$$

Alternatively, we could consider using a 'gating network' to generate data-dependent weights for each model rather than a simple average, though we found the performance to be similar.

Using ensembles to improve the performance of statistical models is a very old idea; see, *e.g.* Opitz & Maclin (1999) for a survey. The basic intuition is that several weak models can be combined in such a way that the ensemble performs better than any individual, and is sometimes explained as being caused by the errors of the models 'cancelling' with one another.

In order to ensure that the models are actually producing different outputs the diversity of the models must be maintained. This can be done in several ways, such as bootstrapping the data, whereby each model gets a slightly different copy of the data, or using totally different model types or architectures. In the case that the model training procedure is convex, and if all models architectures are the same and are getting the same data, then the models in the ensemble would be expected to converge on the same parameters. In the case of neural networks however, the model training procedure is not convex and so our strategy for maintaining diversity is very simple—initialize each model differently. Due to the nature of training neural networks it is likely that differently initialized networks will converge (assuming they do, in fact, converge) to different points of the parameter space. The insight that only different initialization is required is not new, previous papers have observed that different initialization is sufficient for uncertainty estimation Lakshminarayanan et al. (2016); Osband et al. (2016).

Different initialization for networks has an appealing interpretation. If we take a Bayesian approach to the classification problem, then we have a prior over possible model parameters, $p(\theta)$, a likelihood of the data, $p(D|\theta)$, and a probability of a label $y$ given an input and a model, $p(y|x, \theta)$. The 'Bayes-optimal' classification of a new data point $x$ is given by

$$y^\star = \text{argmax}_y \int_\theta p(y|x, \theta)p(D|\theta)p(\theta).$$

This classifier is optimal in the sense that no other classifier can outperform it on average, given the same model class and knowledge of the prior and likelihood; however, the formulation is intractable for all but small problems. We can consider approximating it by the following approach, sample initial parameters from the prior $p(\theta)$ and run an iterative procedure to (approximately) maximize the likelihood $p(D|\theta)$. Very loosely speaking, we can consider this procedure as approximately sampling from the posterior over models $p(\theta|D) \propto p(D|\theta)p(\theta)$. Consequently, we output the classification

$$y^\star = \text{argmax}_y \sum_{i=1}^{k} p(y|x, \theta_i),$$

---

**Algorithm 1** Adversarial ensemble training using PGD under $\ell_\infty$ norm constraint

---

**input:** $k$ neural networks $m_{\theta_i}$, $i = 1, \ldots, k$; attack steps $N$; step sizes $\eta, \hat{\eta}$; initial variance $\sigma$; adversarial loss weighting $\rho$; perturbation width $\delta$
**initialize:** neural network parameters $\theta_i^0$ randomly, $i = 1, \ldots, k$
**for** time-step $t = 0, 1, \ldots,$ **do**
    sample input minibatch $(x, \hat{y}) \sim \mathcal{D}$
    initialize $\tilde{x}^0 = x + \epsilon$ where $\epsilon \sim \mathcal{N}(0, \sigma^2 I)$
    define $\mathcal{B} = \{x' \mid \|x - x'\|_\infty \le \delta\}$
    **for** $k = 0, \ldots, N - 1$ **do**

$$\tilde{x}^{k+1} = \Pi_{\mathcal{B}}(\tilde{x}^k + \hat{\eta} \nabla_x L(\hat{y}, \frac{1}{k} \sum_{i=1}^{k} m_{\theta_i}(\tilde{x}^k)))$$

    **end for**
    update parameters for each $i = 1, \ldots, k$:

$$\theta_i^{t+1} = \theta_i^t - \eta \nabla_{\theta_i} \left( L(\hat{y}, \frac{1}{k} \sum_{j=1}^{k} m_{\theta_j^t}(x)) + \rho L(\hat{y}, \frac{1}{k} \sum_{j=1}^{k} m_{\theta_j^t}(\tilde{x}^N)) \right)$$

**end for**

---

*i.e.*, the best guess of the ensemble. The role of initialization therefore is that of sampling from our prior over possible model parameters.

**Adversarial training of Ensembles** Up to this point we have discussed the use of ensembles for improving classification performance and approximating the Bayes optimal classifier. Typically speaking neural networks appear to not benefit much from ensembling in terms of nominal performance. Here, however, we make the claim that adversarially trained ensembles of networks provide a level of robustness to adversarial attacks. When using ensembles the loss function for adversarial training in (3) is replaced by the mean of the loss over the $k$ models, *i.e.*, now we want to solve

$$\text{minimize} \quad \mathbb{E}_{(x,y) \sim \mathcal{D}} \left( L(\hat{y}, \frac{1}{k} \sum_{i=1}^{k} m_{\theta_i}(x)) + \rho L(\hat{y}, \frac{1}{k} \sum_{i=1}^{k} m_{\theta_i}(\tilde{x})) \right) \tag{4}$$

over variables $\theta_i$, $i = 1, \ldots, k$, and where $\tilde{x}$ is an adversarial example generating by attacking the entire ensemble. The exact procedure is outlined in Algorithm 1. We demonstrate empirically in the numerical results section that this procedure increases robustness to adversarial inputs. Following these results, we offer an analysis and hypothesis why ensembles outperform single models, even when controlling for number of parameters.

## 4 EXPERIMENTAL SETUP

### 4.1 MODELS COMPARED

**Non-Adversarial Benchmarks** The **Baseline** model for our investigation is a Wide ResNet (Zagoruyko & Komodakis, 2016) consisting of a $3 \times 3$ convolution layer, followed by three layers containing 28 ResNet blocks of width factor 10, followed by batch normalization Ioffe & Szegedy (2015) layer, followed by a ReLU (Nair & Hinton, 2010), and by a final linear layer projecting into the logits of the CIFAR-10 classes. All models we experimented with here are variations on this architecture, and where hyperparameters are not explicitly referenced, they are assumed to be the same as this base model. **Ensemble2** contains two copies of the baseline architecture. This has twice the number of parameters of the baseline. Together with the base model, these constitute our non-adversarially trained benchmarks.

**Adversarial Models** When adding adversarial training to the baseline architecture, we obtain our **SingleAdv** benchmark, which has the same number of parameters as the baseline. When trained with adversarial training, whereby the whole ensemble is attacked by Iterated Fast Gradient Sign Method

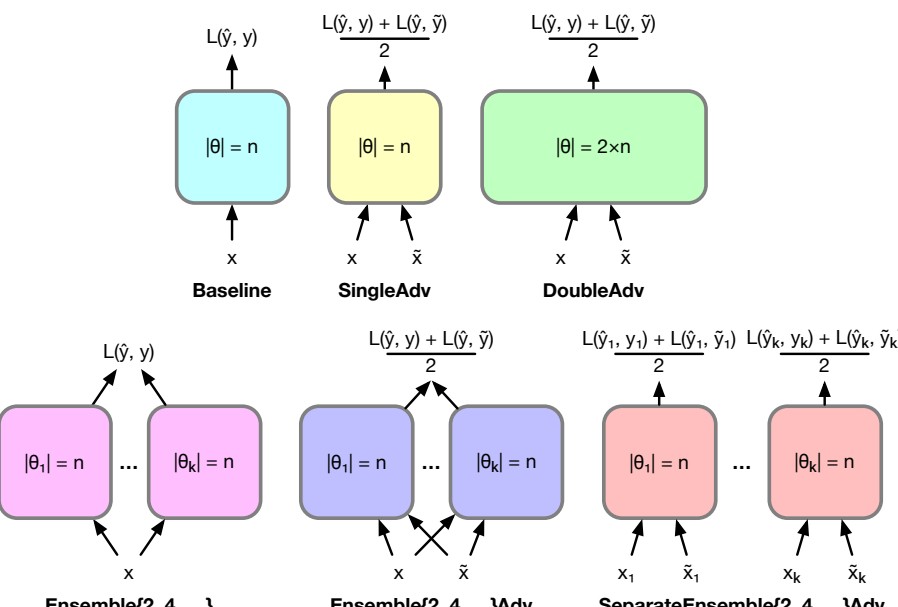

Figure 1: Schematic depiction of classes of models compared in this paper. Here, $n$ indicates the number of parameters in the base model, $\hat{y}$ indicates the ground trouth label, $x$ is a clean input from the dataset, $\tilde{x}$ is that input after a number of steps of the chosen adversarial training attack (7 steps of IFGSM in our experiments), $y$ is the output distribution according to the network based on clean input $x$, and $\tilde{y}$ is the output based on adversarial input $\tilde{x}$. Adversarially trained networks are shown to have two inputs (and two losses) for compactness, but in practice two parameter-sharing copies of the network will be instantiated, with one taking clean input, the other taking adversarial input, and their losses will be computed separately and averaged before optimisation.

(IFGSM) (Kurakin et al., 2016) at each training step to obtain adversarial inputs, we refer to the ensemble as **Ensemble2Adv**. This ensemble has as many parameters as its non-adversarially-trained counterparts.

**Comparisons to Ensemble2Adv**    In order to compare **Ensemble2Adv** to the **SingleAdv** benchmark while controlling for number of parameters, we introduce a variant **DoubleAdv** of this benchmark with ResNet blocks of width 15, which yields roughly the same number of parameters as **Ensemble2Adv**. Finally, we train two separately parameterised instances of **SingleAdv** and ensemble them at test time for the purpose of evaluating the hypothesis that it is adversarial training of ensembles that provides and advantage, and call this test-time model **SeparateEnsemble2Adv**.

The model variations described here are illustrated in Figure 1, which can serve as a basis for repeating these experiments with a different base model architecture.

## 4.2 TRAINING PROCEDURE

We train and evaluate our models on CIFAR-10 (Krizhevsky & Hinton, 2009). We use similar hyperparameters to Zagoruyko & Komodakis (2016), with additional iterations to account for the fact that minimizing the adversarial objective requires more training steps. We train all models for 500,000 iterations using a momentum optimiser with minibatches of size 128, with an initial learning rate of 0.1, a momentum of 0.9, and a learning rate factor (decay) of 0.2 after $\{30k, 60k, 90k\}$ steps. When doing adversarial training, we train both on "clean" versions of the minibatch images, and on adversarial examples produced by 7 steps of IFGSM, following Madry et al. (2017). The cross-entropy losses with regard to the ground truth labels for both the adversarial and clean images are averaged to obtain gradients for the model (*i.e.* $\rho = 1$).

### 4.3 EVALUATION PROCEDURE

During training, we run an evaluation job which evaluates the accuracy of the model on the entire CIFAR-10 test set. We consider two white-box adversaries, both with a maximum $L_\infty$ perturbation of 8 (out of 255): **IFGSM** which performs the iterated fast gradient sign method update, which is equivalent to steepest descent with respect to the $L_{\text{inf}}$ norm Madry et al. (2017); Kurakin et al. (2016) and **PGD** which performs projected gradient descent using the Adam Kingma & Ba (2014) update rule. During training, we evaluate using **IFGSM7**, the training adversary which performs 7 iterations of the IFGSM update, also used in Madry et al. (2017), as well as **PGD5** and **PGD20**, the 5 and 20-step versions of our PGD attack. Additionally, for the best model, we run these attacks 500 steps in order to estimate the strongest possible attacks.

We further include a black-box adversary in our evaluation procedure. We use a dataset of pre-computed adversarial examples, following the procedure in Liu et al. (2016) against an ensemble of a Wide ResNet Zagoruyko & Komodakis (2016) and VGG-like Simonyan & Zisserman (2014) architectures. The two models are trained with standard training procedures and achieve 96.0% and 94.5% accuracy respectively on the CIFAR-10 clean test set, and are ensembled by an arithmetic mean of their logits. The adversary is the **PGD20** adversary which fools all members of the ensemble on 100% of the evaluation set. We note that the exact values for robustness of networks to black box attacks can be highly contingent on the similarity between the original and attacked networks Uesato et al. (2018), rather than the true adversarial robustness of the attacked network. However, we include black box accuracies for best practice, as a check against models which achieve illusory robustness through obscured gradients Goodfellow et al. (2014).

We trained and evaluated each model with three separate random seeds. Evaluation outliers, caused by occasional crashes of evaluation jobs, are removed according to the following procedure. We compute a smoothed version of each time series by using a centered rolling median window of width 50. We take the absolute difference of each original time series and its smoothed form, compute the mean of the difference, and replace points in the original time series with their smoothed version only when the absolute difference exceeds three standard deviations with this mean. This removes at most two outlier points per model per evaluation in our runs. Evaluation time series for different seeds are then interpolated to obtain results on the same 1000 time-steps, which are then averaged across seeds, per model class.

## 5 RESULTS AND ANALYSIS

We give a numerical break down of evaluation accuracies for the metrics described above, during and at the end training, in Table 1: in Table 1a, we report the average of the last 10 evaluation steps for all models, and in Table 1b, we report the evaluation metrics at the time step where each model obtained the best evaluation score on **FGSM5**. In Figures 2a and 2b, we show the evolution of evaluation accuracies for selected metrics. To more thoroughly evaluate the models compared here, we show in Figure 2c how the accuracy of our models drops as the number of PGD attack steps increases. We report the evaluation results for 500 steps of PGD of the model snapshots used for Table 1b in Table 1c.

Figures 2a and 2b show that adversarially trained models uniformly outperform non-adversarially trained ones. Especially with weaker attacks, such as **IFGSM5** and **PGD5**, non-adversarially trained models exhibit some recovery of robustness to attacks after 2–300,000 steps of training, but this is not stable and decays with further training. We further confirm that even such models which achieve some robustness against weak adversaries have true adversarial robustness close to 0% when the adversarial optimization is run for longer. In contrast, the robustness of adversarially trained models is stable throughout training. We read, in Table 1b, that all models incorporating adversarial training do slightly worse on the CIFAR-10 test, suffering a drop of roughly 10 points in accuracy, a phenomenon which was also observed in other work Madry et al. (2017). On **PGD20**, the smallest gap between an adversarially trained model and a baseline is 22%. **Ensemble2Adv** yields an improvement of 7% over a **SingleAdv**, of 5% over the parametetically equivalent **DoubleAdv**, and of 29% over the non-adversarially trained **Ensemble2Adv**.

In Figure 2c, we see that while the accuracies of the **Ensemble2Adv** drop more readily as the number of attack steps increases, they preserve a gap 7 accuracy points over the **SingleAdv** benchmark. Here,

Table 1: Evaluation Results

(a) Average of last 10 evaluation steps

|  | clean accuracy | IFGSM5 accuracy | PGD5 accuracy | PGD20 accuracy | black box accuracy |
|---|---|---|---|---|---|
| **Baseline** | **0.94** | 0.34 | 0.15 | 0.01 | 0.27 |
| **Ensemble2** | **0.94** | 0.59 | 0.44 | 0.30 | 0.22 |
| **Ensemble4** | 0.91 | 0.50 | 0.40 | 0.34 | 0.26 |
| **SingleAdv** | 0.82 | 0.55 | 0.44 | 0.43 | 0.80 |
| **DoubleAdv** | 0.83 | 0.57 | 0.46 | 0.44 | 0.82 |
| **Ensemble2Adv** | 0.85 | 0.62 | 0.55 | 0.52 | 0.83 |
| **Ensemble4Adv** | 0.87 | **0.66** | **0.58** | **0.53** | **0.85** |

(b) Evaluation results for model at best IFGSM5 training step

|  | clean accuracy | FGSM5 accuracy | PGD5 accuracy | PGD20 accuracy | black box accuracy |
|---|---|---|---|---|---|
| **Baseline** | **0.95** | 0.57 | 0.29 | 0.09 | 0.26 |
| **Ensemble2** | **0.95** | 0.65 | 0.52 | 0.38 | 0.22 |
| **Ensemble4** | 0.93 | 0.60 | 0.48 | 0.43 | 0.24 |
| **SingleAdv** | 0.84 | 0.57 | 0.46 | 0.45 | 0.81 |
| **DoubleAdv** | 0.85 | 0.60 | 0.48 | 0.47 | 0.84 |
| **Ensemble2Adv** | 0.87 | 0.64 | 0.56 | **0.52** | 0.85 |
| **Ensemble4Adv** | 0.88 | **0.67** | **0.58** | 0.52 | **0.86** |

(c) Model accuracy after 500 attack steps.

|  | Baseline | SingleAdv | DoubleAdv | Ensemble -2 | Ensemble -2Adv | Separate2Adv |
|---|---|---|---|---|---|---|
| **IFGSM** | 0.16 | 0.46 | 0.47 | 0.13 | **0.55** | 0.49 |
| **PGD** | 0.04 | 0.44 | 0.47 | 0.02 | **0.52** | 0.47 |

we also compare to an ensemble, **Separate2Adv**, where the individual models in the ensemble were separately adversarially trained. We observe that this ensemble produces a robustness to adversarial attacks which is closer to the **SingleAdv** results than to **Ensemble2Adv**, despite having the exact same structure and number of parameters. We present the evaluation accuracies after 500 steps of PGD in Table 1c, which maintains the relative ordering and rough gaps between models seen in Table 1b, thereby helping validate our results.

## 6 DISCUSSION

In this section we briefly discuss the possible reasons for the behaviours observed. As we saw, an ensemble of models trained adversarially outperforms the other setups at test time. We suspect, that this might be happening due to a mechanism described below.

When the model is being trained, it is exposed to pairs of images, both "clean" and adversarially modified. The adversarial training exploits the fact that the original image is close to the decision boundary of the model. The model then, when provided with both clean and adversarial image would attempt to modify the decision boundary in order to engulf them both. It is relatively easy to imagine why **SingleAdv** would be weaker then the other models—it simply has less parameters than the competition. In order to accommodate the adversarial example it has to compromise the decision boundary somewhere else, pulling it close to other clean images, making it vulnerable to subsequent attack. This is illustrated in Figure 3a.

The possible reason why **Ensemble2Adv** outperforms **DoubleAdv** is more elusive. Both models have the same number of parameters, so one could expect them to display a similar performance. As **Ensemble2Adv** is more robust to white box attack during test time we argue, that this might be due to the fact that in abundance of flexibility **DoubleAdv** tends often to spread out thin "tentacles"

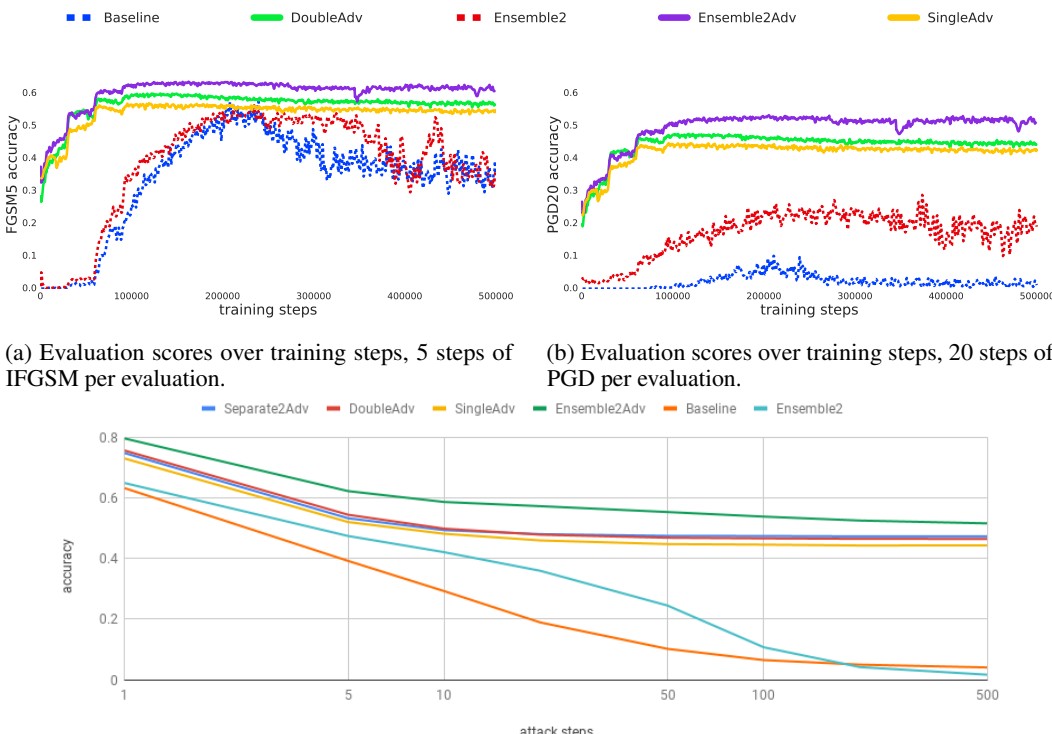

(a) Evaluation scores over training steps, 5 steps of IFGSM per evaluation.

(b) Evaluation scores over training steps, 20 steps of PGD per evaluation.

(c) Accuracy under PGD attack as a function of the number of attack steps.

Figure 2: Evaluation Curves

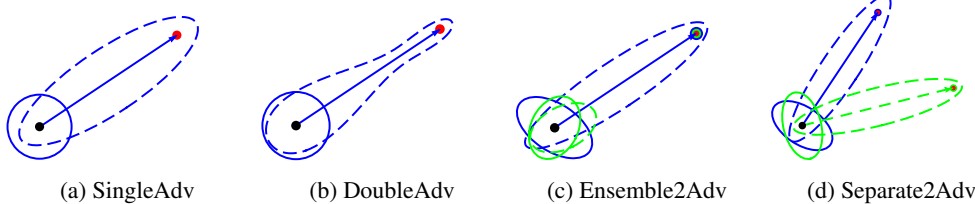

(a) SingleAdv      (b) DoubleAdv      (c) Ensemble2Adv      (d) Separate2Adv

Figure 3: Different responses of various architectures to adversarial training. Solid lines represent decision boundary of the models that see only "clean" images. Dashed lines are the boundaries modified due to the presence of adversarial training. The black dot is a clean image, the red dot is its adversarial modification. In the presence of two models $Model_1$ is blue and $Model_2$ is green. In case it needs to be specified with respect to which model the adversarial example is constructed the red dot has a circle in an appropriate color around it.

(Figure 3b) which do not cover up too much of a space. On the other hand, given that **Ensemble2Adv** is comprised of two separate models they are both subject to lesser ability to overfit following from the smaller number of parameters available in them. Thus we argue, that in most cases the modification of the model with adversarial training covers the adversarial example by modifying one model more than the other. This way the decision boundary of the model modified to a lesser degree still "provides protection" for "clean" images, while at the same time the "tentacle" generated by the model modified more is thicker than the one **DoubleAdv** creates. We illustrate that with Figure 3c.

Finally, it was shown that **SeparateEnsemble2Adv** is outperformed by a **Ensemble2Adv** trained "jointly". We think that this is due to the fact that the adversarial training has to weaken both of the submodels simultaneously. Figure 3d illustrates that.

For a further illustration of the effects of adversarial training we plotted actual images of the decision boundaries for non-adversarially and adversarially trained Baseline and 2-Ensemble models (Figure 4).

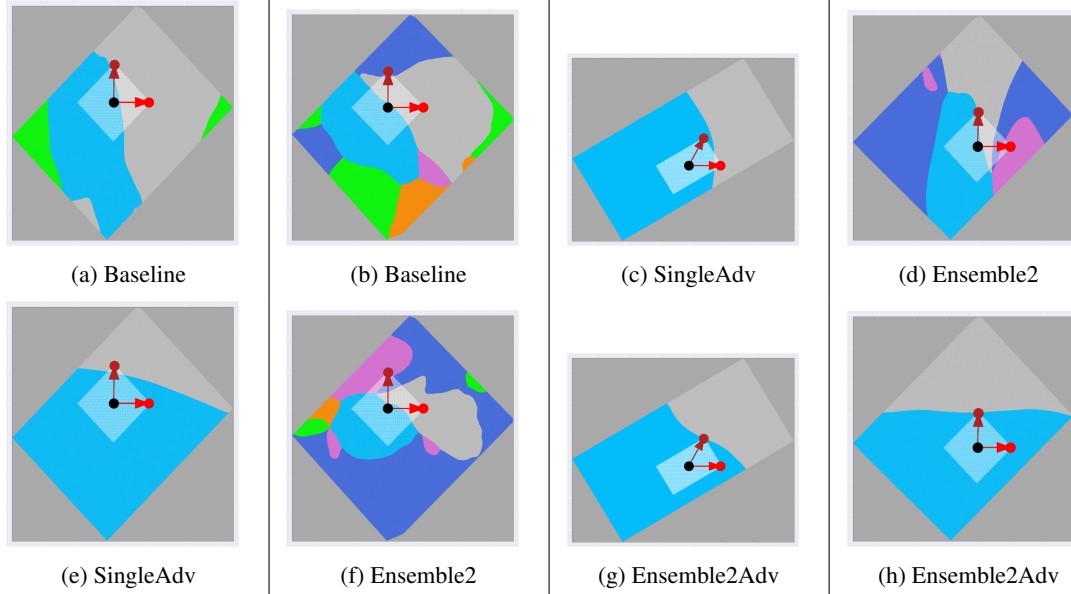

| (a) Baseline | (b) Baseline | (c) SingleAdv | (d) Ensemble2 |

| (e) SingleAdv | (f) Ensemble2 | (g) Ensemble2Adv | (h) Ensemble2Adv |

Figure 4: Decision boundaries for various architectures/training methods. Each column shows the decision regions of two models on the same 2-dimensional plane in the space of all images. On every picture the black dot corresponds to the datapoint—an unaltered (ship) image from the test dataset. The light rectangle superimposed over the dot represents the bounds of the permitted attack region within the region. The two red arrows are the two vectors—attack directions on the base image, with respect to respectively the first and second tested model. The red dots are the images resulting from the attack. The plane presented is then the (unique) 2-dimensional plane containing those 3 points. Dark grey is void (outside the slice boundaries), and all other pixels are generated by a forward pass of the model at those coordinates, with the colour used representing the majority class.

Decision regions of the models are 3072-dimensional sets, so visualizing them itself poses a challenge. What we present are color-coded values of the models restricted to 2-dimensional planes in the space of all images, chosen so that the original image and the closest adversarial example (or attempt to find one) for both models in a pair being compared are co-planar. We observe, amongst other things, some support for the hypothesis put forward in Figure 3: adversarial training adds "thickness" around the natural image points, pushing the boundary further away from them, and in doing so, making adversarial examples harder to find (even within the test set); ensembling makes some classes more "consistent" within the decision plane, but introduce small "pockets" or "tentacles" of other classes; and the combinator thereof removes said pockets to create large regions of the correct class around images. We believe that such an approach of choosing a good plane and plotting the values of models on it is a more informative way of visualizing phenomena taking place in the universe of robustness and adversarial examples than more traditional approaches like t-SNE plots (Maaten & Hinton, 2008).

## 7    CONCLUSIONS AND FURTHER WORK

In this paper, we provide an empirical study of the effect of increasing the number of parameters in a model trained with adversarial training methods, with regard to its robustness to test-time adversarial attacks. We showed that while increasing parameters improves robustness, it is better to do so by ensembling smaller models than by producing one larger model. Through our experiments, we show that this result is not only due to ensembling alone, or to the implicit robustness of an ensemble of adversarially trained models, but specifically to due to the adversarial training of an ensemble as if it were a single model. We proposed a high level interpretation of why this phenomenon might occur. Further work should seek to determine whether scaling the number of models in the ensemble while controlling for number of parameters produces significant improvements over the minimal ensembles studied here in an attempt to draw conclusions about why such architectures are generally more robust than larger single models, even under adversarial training.

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
