# OpenReview forum: "Strength in Numbers: Trading-off Robustness and Computation via Adversarially-Trained Ensembles"
_ICLR.cc/2019/Conference_

### Official Review · AnonReviewer3 · 2018-10-26
**Empirical Study of Robustness of (small) Ensembles of Neural Nets**

**Rating:** 4
**Confidence:** 3

**Review:**

Summary. The paper considers the robustness of neural nets against adversarial attacks. More precisely, the authors experimentally investigate the robustness of ensembles of neural nets. They empirically show that adversarially trained ensembles of 2 neural nets are more robust than ensembles of 2 adversarially trained neural nets.

Pros.
* Robustness of neural nets is a challenging problem of interest for ICLR
* The paper is easy to read
* Experimental results compare different algorithms for 2 neural nets

Cons.
* The study is experimental
* It is limited to gradient-based attacks
* It is limited to ensembles of size 2
* The Ensemble2Adv is a single NN model and not an ensemble model.

Evaluation.
The problem is significant and the use of ensemble methods for robustness against adversarial attacks is a promising line of research. The experimental study in this paper opens new lines of research in this direction. But, in my opinion, the paper is not ready for publication at ICLR. Detailed comments follow but the study is limited to k=2; the main finding is limited to the comparison between bagging two adversarially trained neural nets (SeparateEnsemble2Adv) and learning adversarially the average of two neural nets (Ensemble2adv). In my opinion, Ensemble2adv is a single model of double size and not an ensemble model thus somehow contradicting the main claim of the paper.

Detailed comments.
* Introduction, end of §2, it is said that non-gradient based attacks are still effective. But in the sequel you only consider gradient-based attacks and never discussed this question.
* Introduction, contributions, it should be made clear at the beginning of the paper that you will consider ensembles of size 2 and only gradient-based attacks.
* Section 2. The momentum-based attack should be cited and could be considered. "Boosting adversarial attacks with momentum, Dong et al, CVPR18"
* Section 3, §2, the discussion on ensemble methods is unprecise. Ensemble methods have different objectives. For instance, Bagging-like methods  aim at reducing the generalization error while others as Boosting aim at augmenting the capacity of individual models.
* Section 3. Here is my main concern on this paper. The classical method would be bagging of neural nets with different initializations. The neural nets could be adversarially trained. This would lead to the so-called SeparateEnsemble2Adv. Here, the authors consider another method. Their method can be viewed as k(=2) copies of the same neural network with different initializations and an additional layer computing the average of the k(=2) outputs. Then adversarially learn the obtained model which leads to the so-called Ensemble2Adv algorithm. This algorithm is not an ensemble method as such. In my opinion for k=2, it is equivalent to doubling the size of a neural net, adding averaging of the outputs, and adversarially training the obtained neural net.
* Note recent advances in ensemble NNs with papers such as Averaging weights leads to wider optima ..., Izmailov et al, UAI18; Loss Surfaces, Mode Connectivity, and Fast Ensembling of DNNs, Garipov et al, arXiv:1802.10026
* Section 4.1. Here comes the limitation k=2. The case k=4 is considered in table 1 but is not discussed elsewhere in the paper.
* Section 4.1. I am not convinced by DoubleAdv. It is one way of doubling the size of a neural net but I am not convinced that this is the more efficient. As said before, in my opinion, Ensemble2Adv is another way for doubling the size. And many more should exist.
* Section 5. In my opinion, the main comparisons should concern SeparateEnsemble2Adv and Ensemble2Adv. Also other methods doubling the size should be considered.
* Section 5. For k greater than 2, SeparateEnsemblekAdv should be the better method because the adversarial learning phase could be easily parallelized.
* I am not convinced by the discussion in Section 6.
* Typos. and -> an l-13, p5; IFGSM5, l-19 p6; then l-6 p7; to due l-6 p9
* Biblio. Please give complete references

---

> ### Author Response · Authors · 2018-11-06
> **Thank you for your comments (part 1)**
>
> We thank Reviewer 3 for their detailed comments. We are glad you found the paper to be easy to read, and addressing a problem within an interesting domain. We are sorry to hear you think the paper is not ready for publication. We hope, through this discussion period, to change your mind, or at very least get a better understanding of where you feel the issues with the paper lie and where we can make improvements.
>
> We will respond to your comments in the order we find them.
>
> We first reply regarding your general cons, and will focus on your specific comments in a separate post:
>
> 1) Could you please clarify why the experimental nature of this paper is a con? We agree that it would be a strong plus to have a strong theoretical understanding of why the phenomenon observed in our experiments occurs, and we have attempted to discuss this in Section 6, but we would would argue that a sound theoretical understanding of adversarial attacks and robustness is lacking throughout the literature, perhaps due to this subfield being very young, and are unsure what is lacking in our paper that is present in others. Our field is primarily empirical, and we have attempted to be thorough in our study of the effect of ensembles with regard to robustness to adversarial examples.
>
> 2) As we point out to R2, Madry (2017) and the appendix of Uesato (2018) argue that gradient-based attacks are ubiquitously more successful and more adapted to models which do not defensively obfuscate gradients. Adversarially trained models, a category under which the models we explore in this paper (except for the simplest baselines) fit, do not obfuscate gradients, and thus gradient based attacks (and robustness against them) are the most reasonable focus for this work.
>
> 3) We have results for ensembles of size 4 as well. See Tables 1(a) and 1(b). That said, the main focus of the study is the effect of adversarially trained ensembles rather than how robustness scales with the size of the ensemble, and to this end, what really matters is that the phenomenon is reliably attested when adversarially training even the smallest ensemble vs. a single model (when controlling for number of parameters, hyperparameters, and training mechanisms), and that is what our ablation study focusses on.
>
> 4) The output is the average of component model probabilities, and as such decomposes into two separate losses (since grad distributes across addition). The only way in which the models are couples is that the adversarial examples fed to them during adversarial training are produced by treating the ensemble as a single model, but once the examples are produced (and for clean training), it does not matter that the models are trained on different machines, different optimisers, etc. In that sense, it is a true (albeit very simple) ensemble, but if you do not agree we would be grateful to receive an explanation as to why it is not a proper ensemble.
>
> References:
> Aleksander Madry, Aleksandar Makelov, Ludwig Schmidt, Dimitris Tsipras, and Adrian Vladu. Towards deep learning models resistant to adversarial attacks. arXiv preprint arXiv:1706.06083, 2017.
> Jonathan Uesato, Brendan O’Donoghue, Aaron van den Oord, and Pushmeet Kohli. Adversarial risk and the dangers of evaluating against weak attacks. In The 35th International Conference on Machine Learning (ICML), 2018.

---

> ### Author Response · Authors · 2018-11-06
> **Thank you for your comments (part 2)**
>
> > Introduction, end of §2, it is said that non-gradient based attacks are still effective. But in the sequel you only consider gradient-based attacks and never discussed this question.
>
> As we discussed in part 1, gradient based attacks are the most natural attack to consider here, but we will include this rationale in the introduction so as to not confuse readers.
>
> > Introduction, contributions, it should be made clear at the beginning of the paper that you will consider ensembles of size 2 and only gradient-based attacks.
>
> We can certainly make it clearer. Thanks for the suggestion.
>
> > Section 2. The momentum-based attack should be cited and could be considered. "Boosting adversarial attacks with momentum, Dong et al, CVPR18"
>
> Thanks for the suggestions. We will look into it and cite this as related work.
>
> > Section 3, §2, the discussion on ensemble methods is unprecise. Ensemble methods have different objectives. For instance, Bagging-like methods  aim at reducing the generalization error while others as Boosting aim at augmenting the capacity of individual models.
>
> We appreciate that there are a variety of ways to do ensembles, which offer significant benefits. The fact that our results are attested with a very simple, very naive form of ensemble (just averaging predictions) should be taken as a strength of the paper rather than a limitation thereof, as we ablate away other factors such as whether we are bagging, boosting, etc.
>
> > Section 3. Here is my main concern on this paper. The classical method would be bagging of neural nets with different initializations. The neural nets could be adversarially trained. This would lead to the so-called SeparateEnsemble2Adv. Here, the authors consider another method. Their method can be viewed as k(=2) copies of the same neural network with different initializations and an additional layer computing the average of the k(=2) outputs. Then adversarially learn the obtained model which leads to the so-called Ensemble2Adv algorithm. This algorithm is not an ensemble method as such. In my opinion for k=2, it is equivalent to doubling the size of a neural net, adding averaging of the outputs, and adversarially training the obtained neural net.
>
> As discussed above and in part 1, the model is a very simple, naive form of assembling, and this is done by design. We compared to, and controlled for, models which are separately trained, single models which have as many parameters, ensembles of (separately) adversarially trained ensembles, and showed that none of them beat an equivalent (in terms of number of parameters) model with shared adversarial examples and adversarial training. As an aside, it is unclear to us that a collection of models is not ensemble by virtue of not having been trained on the same data. For models with convex error surfaces this is obviously required (else the models collapse onto the same parameters) but for DNNs such as WideResNets this does not happen due to the proliferation of local minima.
>
> > Note recent advances in ensemble NNs with papers such as Averaging weights leads to wider optima ..., Izmailov et al, UAI18; Loss Surfaces, Mode Connectivity, and Fast Ensembling of DNNs, Garipov et al, arXiv:1802.10026
>
> Thank you for the suggested readings. We have no doubt that more sophisticated approaches to ensembles my provide even better results. The focus here was to demonstrate that the phenomenon, which our experiments show is due to the production of shared adversarial examples (i.e. those which attack the ensemble as a whole) during adversarial training, is exhibited even for the simplest ensembles.
>
> > Section 4.1. Here comes the limitation k=2. The case k=4 is considered in table 1 but is not discussed elsewhere in the paper.
>
> As discussed above, the focus of this paper was primarily to show that the production of shared adversarial examples for ensembles beats non-shared examples for ensembles during adversarial training, or adversarial training of single models. In this sense, it seems most reasonable to show the results for the smallest possible ensemble. We provided results for Ensembles of size 4 in Table 1 just to give an idea of how the results vary with scale (the gap is small and the returns are diminishing), but this is not the focus of the paper.
>
> > Section 4.1. I am not convinced by DoubleAdv. It is one way of doubling the size of a neural net but I am not convinced that this is the more efficient. As said before, in my opinion, Ensemble2Adv is another way for doubling the size. And many more should exist.
>
> There are different ways of controlling for number of parameters, as you suggest. We primarily looked at DoubleAdv and SeparateEnsemble2Adv as equivalent (in terms of the number of parameters) models to Ensemble2Adv because there are just not that many other alternatives for WideResNets (while maintaining topologically similar models), but we recognise that for other architectures, other alternatives may exist.

---

> ### Author Response · Authors · 2018-11-06
> **Thank you for your comments (part 3)**
>
> > Section 5. In my opinion, the main comparisons should concern SeparateEnsemble2Adv and Ensemble2Adv. Also other methods doubling the size should be considered.
>
> We did consider other methods which control for the number of parameters within a single model. See DoubleAdv.
>
> > Section 5. For k greater than 2, SeparateEnsemblekAdv should be the better method because the adversarial learning phase could be easily parallelized.
>
> One of the key results of our ablation study was precisely to observe that this is not the case: it is the production of *shared* adversarial examples (obtained by treating the ensemble as a whole when producing an adversarial example for adversarial training) which provides the difference in results between SeparateEnsemble2Adv and Ensemble2Adv. As stated in part 1, once the shared adversarial example is produced (or during the non-adversarial training component of overall training), the component models of Ensemble2Adv can be trained in parallel just like SeparateEnsemble2Adv. The *only* difference is in the production of the adversarial examples.
>
> > I am not convinced by the discussion in Section 6.
>
> We are sorry to hear this. The method of visualisation we developed to attempt to study this phenomenon is–to the best of our knowledge–new, and we would be grateful to hear which aspect of our discussion and analysis you found unconvincing.
>
> > Typos. and -> an l-13, p5; IFGSM5, l-19 p6; then l-6 p7; to due l-6 p9
>
> Will fix. Thanks.
>
> > Biblio. Please give complete references
>
> Will fix. Thanks.

---

### Official Review · AnonReviewer2 · 2018-10-31
**Simple but effective variant on training ensemble of independently adversarially trained models**

**Rating:** 6
**Confidence:** 3

**Review:**

The paper proposes to train an ensemble of models jointly, where the coupling lies in that at each time step, a set of examples that are adversarial for the ensemble itself is incorporated in the learning.

The experiments are thorough and compare multiple types of attacks, although they are all based on gradients (while the paper does mention recent attacks that do not rely on gradients so much). The results are rather convincing and show a clear difference between the proposed method and independently training the models of the ensemble (even if each one is training with examples adversarial to itself).

The paper is clear and well-written.

Pros:
- The superior performance of the proposed method
- The method is simple and thus could have a practical impact
- Clear and thorough analysis

Cons:
- Only gradient based attacks (which are somewhat criticized in the introduction)
- Novelty may be a bit limited: this is a rather small variation on existing stuff (but it works rather well)

Remarks:
- Fig 2c could use the same line styles and order as Fig 2a/2b
- "a gap 7 accuracy"?

---

> ### Author Response · Authors · 2018-11-06
> **Thank you for your comments**
>
> We thank Reviewer 2 for their comments. We are happy to hear the paper was found to be clear and well written, with thorough analysis.
>
> Regarding the objections made, we are hoping we can obtain a bit more detail about how you think these impact the paper.
>
> 1) Madry (2017) and the appendix of Uesato (2018) argue that gradient-based attacks are ubiquitously more successful and more adapted to models which do not defensively obfuscate gradients. Adversarially trained models, a category under which the models we explore in this paper (except for the simplest baselines) fit, do not obfuscate gradients, and thus gradient based attacks (and robustness against them) are the most reasonable focus for this work. That said, while slightly outside the scope of our intended study, it would be interesting to see how gradient-free methods fare against adversarially trained ensembles. We will need to reserve this for further work, as it is not possible to train adversarially trained ensembles to completion by the end of the rebuttal period, but we welcome your suggestion for a follow-on study.
>
> 2) We are unsure where the lack of novelty lies. We appreciate the method is simple and general, but we are unaware of any similar study to that which we provide through our broad ablative evaluation of this phenomenon. Could the reviewer kindly clarify specify in what way we could have improved the "novelty" aspect of our work, and perhaps point us to the existing work this is perceived as being a variation thereof?
>
> A further thank you for your remarks and spotting a typo. We will fix.
>
> References:
> Aleksander Madry, Aleksandar Makelov, Ludwig Schmidt, Dimitris Tsipras, and Adrian Vladu. Towards deep learning models resistant to adversarial attacks. arXiv preprint arXiv:1706.06083, 2017.
> Jonathan Uesato, Brendan O’Donoghue, Aaron van den Oord, and Pushmeet Kohli. Adversarial risk and the dangers of evaluating against weak attacks. In The 35th International Conference on Machine Learning (ICML), 2018.

---

### Official Review · AnonReviewer1 · 2018-11-01
**Possibly significant result but requires more experimental analysis**

**Rating:** 5
**Confidence:** 4

**Review:**

This paper presents a new adversarial training defense whereby an ensemble of models is trained against both benign and adversarial examples. The authors demonstrate on the CIFAR-10 dataset that the ensemble has improved robustness against a wide variety of white-box and transfer-based black-box attacks compared to other adversarial training techniques. The results appear significant but would greatly benefit from more thorough experiments.

Pros:
- Conceptually simple and intuitive.
- Thorough baselines and attack methods.

Cons:
- Limited novelty.
- Needs more experimental validation against other datasets (e.g. ImageNet) and models (e.g. Inception-v3).
- Table 1 shows that the clean accuracy of adversarially trained models is significantly worse, which suggests that some aspect of training was done improperly.

-----------------------------

I would like to clarify regarding the listed cons:

- Limited novelty: Adversarial training has been well-established as a viable defense against adversarial examples, as well as training a single model against an ensemble of adversarial examples crafted on different networks (Tramer et al. https://arxiv.org/pdf/1705.07204.pdf). While this work is sufficiently different from prior methods, its novelty is insignificant.
- More experimental validation: Due to the limited novelty, it is crucial that the authors validate their result more thoroughly to eliminate any doubt on applicability, especially against more challenging datasets such as ImageNet. While the experiments on CIFAR-10 are certainly sufficient, it is dangerous, particularly in works on defenses against adversarial examples, to restrict only to a relatively simple dataset.
- Tramer et al. (https://arxiv.org/pdf/1705.07204.pdf) publicly released their ensemble adversarially trained Inception-v3 model that has the same top-1 and top-5 clean accuracy on ImageNet as the base model. This serves as evidence that it is certainly possible to adversarially train a model without compromise to clean accuracy, especially on a simpler dataset such as CIFAR-10.

---

> ### Author Response · Authors · 2018-11-06
> **Thank you for your comments**
>
> We thank Reviewer 1 for their comments, and are happy that they found the paper thorough and intuitive.
>
> We would like to discuss the cons outlines in their short review, in the hope of clarifying the rationale behind the rating provided.
>
> 1) Can the reviewer please clarify what they mean by "limited novelty"? We are unable to offer a counter-argument without more detail as to why the reviewer does not find the paper/method novel.
>
> 2) We agree that more evaluation is always a good thing, but time and resources do not always permit it. Could the reviewer clarify why the fairly extensive longitudinal study provided in our experiments is insufficient to convince them of the results?
>
> 3) All papers that we are aware of reporting clean accuracy for adversarially trained models show a similar drop with regard to the non-adversarially trained models (with same architecture, hyperparameters, etc). See e.g. Madry (2017) for details:
>
> References:
> Aleksander Madry, Aleksandar Makelov, Ludwig Schmidt, Dimitris Tsipras, and Adrian Vladu. Towards deep learning models resistant to adversarial attacks. arXiv preprint arXiv:1706.06083, 2017.

---

### Meta-Review · Area_Chair1 · 2018-12-17
**Reject**

**Confidence:** 5
**Recommendation:** Reject

**Metareview:**

The work brings little novelty compared to existing literature.